# Dietary and Nutritional Interventions in Nonalcoholic Fatty Liver Disease in Pediatrics

**DOI:** 10.3390/nu15224829

**Published:** 2023-11-18

**Authors:** Camila Farías, Camila Cisternas, Juan Cristobal Gana, Gigliola Alberti, Francisca Echeverría, Luis A. Videla, Lorena Mercado, Yasna Muñoz, Rodrigo Valenzuela

**Affiliations:** 1Department of Nutrition, Faculty of Medicine, Universidad de Chile, Santiago 8380453, Chile; camila.farias.c@uchile.cl (C.F.); cpcisternas@uc.cl (C.C.); lorena.mercado@gmail.com (L.M.); yasnamunoz.nut@gmail.com (Y.M.); 2Department of Pediatric Gastroenterology and Nutrition, Division of Pediatrics, School of Medicine, Pontificia Universidad Católica de Chile, Santiago 8330023, Chile; jcgana@gmail.com (J.C.G.); gigialberti@gmail.com (G.A.); 3Nutrition and Dietetic School, Department of Health Sciences, Pontificia Universidad Católica de Chile, Santiago 8331150, Chile; franciscaecheverria@uc.cl; 4Molecular and Clinical Pharmacology Program, Institute of Biomedical Sciences, Faculty of Medicine, University of Chile, Santiago 8380000, Chile; lvidela1944@gmail.com; 5Escuela de Nutrición y Dietética, Facultad de Farmacia, Universidad de Valparaíso, Valparaíso 2360134, Chile

**Keywords:** pediatric NAFLD, exercise, eating patterns, polyunsaturated fatty acids, vitamins

## Abstract

Nonalcoholic fatty liver disease (NAFLD) is pediatrics’ most common chronic liver disease. The incidence is high in children and adolescents with obesity, which is associated with an increased risk of disease progression. Currently, there is no effective drug therapy in pediatrics; therefore, lifestyle interventions remain the first line of treatment. This review aims to present an updated compilation of the scientific evidence for treating this pathology, including lifestyle modifications, such as exercise and dietary changes, highlighting specific nutritional strategies. The bibliographic review was carried out in different databases, including studies within the pediatric population where dietary and/or nutritional interventions were used to treat NAFLD. Main interventions include diets low in carbohydrates, free sugars, fructose, and lipids, in addition to healthy eating patterns and possible nutritional interventions with *n*-3 polyunsaturated fatty acids (EPA and DHA), amino acids (cysteine, L-carnitine), cysteamine, vitamins, and probiotics (one strain or multi-strain). Lifestyle changes remain the main recommendation for children with NAFLD. Nevertheless, more studies are required to elucidate the effectiveness of specific nutrients and bioactive compounds in this population.

## 1. Introduction

The increase in overweight and obesity prevalence has affected the entire population worldwide, increasing the risk of chronic non-communicable diseases at an early age. Among the pathologies caused by this situation, nonalcoholic fatty liver disease (NAFLD) stands out [1]. Anderson et al. conducted a systematic review and meta-analysis to estimate the prevalence of NAFLD in children and adolescents up to 19 years old, using data from nine general population studies up to 2013. The authors estimated a prevalence of 2.3% in children with normal weight, 12.5% in those overweight, and 27.5% and 36.8% in individuals with obesity between 5 and 15 years old and over 15 years old, respectively [2]. Subsequently, a study by Li et al. estimated that the prevalence of NAFLD in the pediatric age group is 7.4% and that in children with obesity, the prevalence reaches 52.5%, regardless of the diagnostic method, projecting a prevalence of 30.7% for the year 2040 [3]. Added to these studies, the prevalence of NAFLD in populations with obesity varies geographically with a higher prevalence being reported in Central America and the Middle East, with 42.5% in children aged 8–11 years determined by alanine aminotransferase (ALT) levels in blood serum [4,5].

The pathophysiology of NAFLD in childhood or adolescence is complex and is related to hereditary and environmental factors. Studies have shown that if parents have hepatic steatosis or diabetes mellitus, their children are at significantly increased risk of developing NAFLD [6]. The relation between weight and height, defined as the body mass index (BMI), is one of this pathology’s most critical risk factors. In this regard, prospective studies in children show a clear association between increased BMI in childhood and a greater risk of developing NAFLD in adolescence and adulthood [4,7]. Importantly, several conditions have been estimated to enhance the risk of overweight in childhood, including the COVID-19 pandemic, the global economic crisis [8], or natural disasters [9].

It has been observed that the consumption of sugar added to foods and beverages as part of the processing or preparation (added sugars) accounts for 13–17% of calories in the diet of American children and adolescents, according to the 2009–2012 National Health and Nutrition Examination Survey [10]. Based on all of the above, it is emphasized that free sugar consumption has a role in promoting insulin resistance (IR), hyperuricemia, and the development of NAFLD [11], and within the postnatal risk factors for NAFLD in children is the high consumption of fructose and other carbohydrates [10,12]. In this context, many research groups seek to establish the relationship between free sugar and fructose intake and the development of obesity and NAFLD in the pediatric population [13]. In addition, the quality of fat in the diet plays a relevant role in the development of NAFLD, which has been reported in various studies [3,14,15,16]. Evidence suggests that consumption of an isocaloric diet rich in saturated fatty acids can contribute to hepatic fat accumulation, in contrast to a diet rich in polyunsaturated fatty acids (PUFA)—particularly, *n*-3 fatty acids [17]. Given the pathophysiology, it is possible that different diets may have a role in the development of NAFLD (Figure 1). In this context, the Expert Committee on NAFLD (ECON) and the North American Society for Pediatric Gastroenterology, Hepatology, and Nutrition (NASPGHAN) recommend lifestyle modifications to improve diet quality and increase physical activity as first-line treatment in pediatric patients with NAFLD [18]. Nevertheless, research continues to find new treatments focused on the pediatric population due to the lack of consistent evidence. The new lines of investigation contemplate modifications in the distribution of macronutrients, evaluation of eating patterns, and/or addition of micronutrients and probiotics. Considering these antecedents, this review presents an updated account of the dietary treatments of NAFLD in pediatrics that involve lifestyle interventions, including exercise, dietary, and/or nutritional modifications (Figure 2).

## 2. Methods

To select and identify eligible studies, the databases PubMed, Web of Science, and the library catalog of the University of Chile were searched using the following search terms: “Pediatric NAFLD”, “NAFLD in children”, “MAFLD in children”, or “Pediatric NAFLD Nutrition”. For the purposes of this review the articles selected were experimental and observational studies (case-control and cross-sectional study) looking at dietary and/or nutritional interventions or dietary behavioral assessments for the treatment of NAFLD within the pediatric population (3 to 18 years of age) published between 2006 and 2023 in English and Spanish. To improve the review process and select and identify additional eligible studies, the Scale for the Assessment of Narrative Review Articles (SANRA) [19] was applied, excluding studies that did not reach a minimum of six points. Of 102 abstracts initially selected, only 28 met the search and selection criteria.

## 3. Lifestyle Modifications

In relation to the increased incidence of NAFLD in the pediatric population and its possible consequences, the search for treatment has become the subject of numerous scientific studies [3]. According to the NASPGHAN and American Association for the Study of Liver Diseases (AASLD) guidelines, lifestyle modification remains the main method of treatment [20]. Lifestyle modifications include teaching children to adopt a healthy lifestyle and diet, as well as encouraging them to include a scheduled and supervised physical activity program [21].

### 3.1. Exercise and Dietary Changes

There are several randomized controlled clinical trials (RCTs) in which the impact of the modification of the diet in children with NAFLD was evaluated [22], emphasizing the children’s food education, since, through knowledge, they can access and choose healthier options (Table 1). In this area, Chan et al. conducted an RCT on 52 Chinese adolescents with obesity, aiming to expand knowledge and modify behavior in terms of food choice [23]. The authors evaluated the effect of a lifestyle intervention, counseling the participants initially once a week for four months on diet and exercise, followed by a maintenance phase, including visits twice a month for 52 weeks. The intervention reduced body fat and intrahepatic triglycerides (TG) content measured by proton magnetic resonance spectroscopy [23]. Malecki et al. studied 49 patients between 3 and 16 years old diagnosed with NAFLD by ultrasound, who were recommended to follow a Mediterranean diet and an aerobic physical exercise routine of moderate intensity for at least 60 min and five days a week. In addition, they were limited to using electronic devices (television, cell phones, tablets, among others) for less than 2 h a day, and the patients were followed for 2.45 ± 1.45 years [20]. The main result was a decrease in aspartate aminotransferase (AST) and ALT levels in all patients, including those who did not diminish their BMI [20]. In another study by Henning et al., 117 children with obesity, 43% of whom had NAFLD and 50% who showed elevated aminotransferases and decreased insulin sensitivity, were followed for 10 weeks [21]. The treatment consisted of a hypocaloric diet that included three meals daily (breakfast, lunch, and dinner) with three fruit or vegetable snacks and exercising for 1 h a day. Data reported showed a decrease in body weight, an increase in insulin sensitivity, and an improvement within NAFLD parameters such as liver echogenicity and texture, decreased ALT, alkaline phosphatases, γ-glutamyltransferase, and coagulation factors II, VII, and X [21]. 

### 3.2. Exercise Interventions

Regarding exercise as a part of lifestyle modifications, the RCT by De Piano et al. compared the effect of aerobic exercise with aerobic exercise plus resistance on NAFLD in 68 post-pubertal obese adolescents. The authors found a greater efficacy in the group that followed the aerobic and resistance exercise protocol. This group exhibited lower insulin levels, insulin resistance index (HOMA-IR), and ALT compared with the aerobic-exercise-only group [24]. Subsequently, De Lira et al. evaluated the effect of low- or high-intensity training in adolescents with obesity and NAFLD, finding an improvement in the disease biomarkers of both groups, decreasing AST and ALT, and increasing the levels of high-density lipoproteins (HDLs) [25]. Iraji et al. compared the effect of school exercise and high-intensity interval training in 34 male adolescents with diagnosed NAFLD and obesity. The authors reported that all participants significantly decreased body fat percentage, IR, TG levels, total cholesterol, ALT, and AST, pointing towards an effective improvement in markers related to NAFLD [26]. On the other hand, to evaluate the required time of the intervention, Kang et al. assessed the effect of intense physical activity in a short period of 7 days on liver fat content by abdominal computed tomography. In children with obesity, the brief intervention resulted in acute weight loss with lower fat mass and serum insulin. However, liver fat content increased independent of IR [27]. Given these results, ECON and NASPGHAN recommended increasing moderate- to high-intensity physical activity and limiting screen time activities to <2 h per day for all children, including those with NAFLD [19] (Table 2). 

## 4. Specific Dietary Interventions

### 4.1. Carbohydrates

Carbohydrates usually provide between 55 and 75% of daily energy requirements. Although they are not essential nutrients, they substantially impact health maintenance [18]. Therefore, among the various dietary options, limiting the intake of carbohydrates is an easily approachable option for treating NAFLD in children (Table 3). 

#### 4.1.1. Low Carbohydrate Diets

Animal and human evidence support the adverse effects of high sucrose and glucose intake, particularly fructose consumption, on obesity and pediatric NAFLD risk. Additional glucose promotes de novo lipogenesis (DNL) by activating the transcription factor of the carbohydrate-response element-binding protein (CHREBP). In addition, DNL has been shown to be abnormally dysregulated in NAFLD patients [21,28]. In this context, Goss et al. evaluated the effects of a low-carbohydrate diet through an RCT on 32 participants from 9 to 18 years old with obesity and NAFLD who received a diet restricted in carbohydrates (<25% carbohydrate, >50% fat, 25% protein), or a fat-restricted control diet (55% carbohydrate, 20% fat, 25% protein) plus dietary education for eight weeks [14]. The restricted carbohydrate diet was found to improve body composition by inducing a decrease in abdominal fat mass and body fat mass compared to the control group. In addition, carbohydrate restriction significantly reduced fasting insulin and IR, and liver lipids assessed by MRI significantly diminished (−32%) only within the group with carbohydrate restriction [14]. Nonetheless, the authors indicate that the sample size may not be large enough to determine whether this decrease in liver fat was due to a unique effect of carbohydrate restriction [14]. It is important to note that these studies are short-term. A long-term low-carbohydrate diet is difficult to adhere to and could interfere with the recommended intake of fruits and cereals for the pediatric population, which could eventually have negative consequences on development.

#### 4.1.2. Low-Free-Sugars Diets

Free sugars are monosaccharides and disaccharides, which can be naturally present or added to foods/beverages by the manufacturer or consumer (added sugars) [29]. The current guidelines of the World Health Organization recommend that the daily intake of free sugars should be limited to less than 10% for all people and less than 5% for specific circumstances. This recommendation is for added sugars, not intrinsic or natural ones in fresh fruits and vegetables [17,29].

Schwimmer et al. published an RCT that included adolescents from 11 to 16 years old with NAFLD who followed a diet low in free sugars (<3% of the calories from free sugar) plus nutritional education for eight weeks [18]. They observed a significant reduction in hepatic steatosis (from 17% to 25% diminution) estimated by magnetic resonance imaging-proton density fat fraction in the treated group. However, it is emphasized that the intervention did not reduce hepatic steatosis or ALT levels to the normal range; therefore, longer-term trials were suggested [17]. Subsequently, Cohen et al. [30] carried out a sub-study considering 29 adolescents with NAFLD who participated in the RCT by Schwimmer et al. [17]. At the end of the study, the authors reported that hepatic DNL (measured as percentage contribution to plasma triglyceride palmitate using a 7-day metabolic labeling protocol) was significantly lower in the treated group (24.1% to 34.6% reduction) versus the control group (33.9% to 34.6% decrease). Moreover, the adolescents who received a low-sugar diet exhibited decreased fasting insulin and correlations between liver DNL, fasting insulin, and high free sugar intake [30].

#### 4.1.3. Low Fructose Diets

Fructose is a non-essential nutrient that can be synthesized from glucose via the polyol pathway in humans. This free sugar is found in honey (50% fructose), in sweeteners such as high-fructose corn syrup (42% or 55% fructose), in fruit juice, and is also found along with glucose in the form of sucrose, commonly known as table sugar [11]. After fructose ingestion, this monosaccharide is rapidly absorbed into enterocytes. Fructose enters the liver, where it is phosphorylated and metabolized into glucose, fatty acids, and triglycerides [11,15]. Unlike glucose, fructose is more easily converted into fatty acids due to an increase in the expression of lipogenic enzymes and their exclusive metabolization in the liver [15]. On the other hand, the phosphorylation of fructose uses ATP, which leads to ATP depletion that induces ATP deaminase activity, favoring uric acid production and TG accumulation [31]. 

In this respect, Jin et al. sought to test whether hepatic steatosis and associated cardiovascular risk factors would improve after replacing high-fructose beverages with glucose-only drinks [13]. This study included overweight Hispanic American adolescents with NAFLD, divided into two groups, who were given three daily servings of 237 mL beverages (33 g of carbohydrates from fructose or glucose) for four weeks. Both groups had no changes in liver fat or body weight, probably due to the short intervention period. However, in the group of drinks with glucose, there was a significant improvement in factors related to cardiovascular disease (insulin sensitivity, high-sensitivity C-reactive protein, and low-density lipoprotein (LDL) oxidation) [13]. On the other hand, O’Sullivan et al. evaluated the associations between fructose and total sugar intake and post-NAFLD diagnosis in obese and non-obese adolescents in a longitudinal birth cohort at ages 14 and 17 [15]. They observed that after energy adjustment in multivariable logistic regression models, adolescents with obesity had a higher intake of fructose, which was associated with a significant increase in the risk of developing NAFLD (specifically, for each 1 g increase in fructose, the probability of developing the disease increases by 9%) [15]. Later, Sullivan et al. measured fructose absorption/metabolism in pediatric NAFLD compared to obese and lean controls in a pilot study [31]. The groups received oral fructose (1 g/kg ideal body weight), and following fructose ingestion, NAFLD children had elevated serum glucose, insulin, and uric acid levels but lower fructose excretion and breath hydrogen levels (test to evaluate fructose malabsorption) compared to lean subjects. In contrast, obese children without NAFLD showed an intermediate response. Thus, children with NAFLD seem to have greater absorption and an exaggerated metabolic response to fructose administration compared to lean children, which could contribute to their pathophysiology. However, whether this may be related to the upregulation of GLUT5 and fructokinase by prior fructose exposure or due to genetic/ethnic differences was not determined [31]. Moreover, a cross-sectional study by Hamza et al. evaluated the intake of fructose in the diet of 55 obese children and adolescents compared with 30 healthy non-obese children aged 6 to 14. Based on the data reported, the researchers noted that dietary fructose intake from processed sources and energy intake were significantly higher in patients with obesity. Furthermore, they detected NAFLD in 50 cases of obese children and verified the association between dietary intake of fructose and NAFLD development [32]. However, more research is still required considering a larger sample to elucidate the protective role of a low fructose diet in developing NAFLD in adolescents with obesity.

### 4.2. Fatty Acids

One of the aims of the study by Gibson et al. was to examine whether excessive or deficient intakes of specific nutrients differ between children with NAFLD and obese children without evidence of liver disease [16]. They found that NAFLD subjects consistently had lower PUFA intake than controls, while saturated fatty acid intake appeared higher. However, after applying the multiple hypothesis testing correction, there were no significant differences in macronutrient and micronutrient intake between the NAFLD and the obese group [16].

#### Low-Fat Diets

There is little evidence of dietary interventions in pediatrics that involve a low-fat diet for the treatment of NAFLD; in our search, we identified two studies. In the first one conducted by Goss et al., it was reported that in children with NAFLD, a diet low in carbohydrates and sugars for eight weeks is effective in reducing intrahepatic fat accumulation, IR, and visceral fat compared to a fat-restricted diet [11]. In contrast, Yurtdas et al. evaluated a low-fat diet on hepatic steatosis, inflammation, and oxidative stress in adolescents with obesity and NAFLD for a more extended period (12 weeks). The low-fat diet was 50–60% carbohydrate, <30% fat (with <10% saturated fat), and 20% protein, and participants received dietary recommendations [33]. Data reported an improvement in anthropometric parameters and body composition, and the subjects reduced liver steatosis, liver enzymes, insulin levels, HOMA, and IL-6 [30]. It should be considered that changes in diet composition differed between groups as predicted after the dietary intervention. Subsequently, Akbulut et al. evaluated the effects of a low-fat diet in reducing hepatic steatosis in children and adolescents aged 9 to 17 with NAFLD for 12 weeks. The low-fat diet was normal caloric and consisted of 55% carbohydrate, 20–25% fat, with <10% energy from saturated fat, and 20–25% proteins, and contemplated nutritional education [31]. A low-fat diet contributed to a decrease in hepatic steatosis and an improvement in IR [12]. These results show that a fat-restricted diet in treating pediatric NAFLD is possible in the short term (minimum 12 weeks). In addition, it is noteworthy that the diet restricted in fatty acids proposed by both studies [11,31] considers the lower limit of the recommended dietary intake of total fat (25 to 35% of total caloric energy) for children and adolescents from 2 to 18 years of age recommended by FAO. This is similar to saturated fatty acids, whose reduction was near to the recommendation (less than 8% of the total caloric energy) [32].

### 4.3. Dietary Patterns

It is necessary to identify the dietary characteristics of populations, describing the groups of foods and nutrients consumed and their combination, variety, frequency, and quantity of food since the correct identification allows for more precise recommendations. For this reason, dietary patterns have become relevant in public health [33,34] and could be a crucial issue to address in pediatric NAFLD.

#### 4.3.1. Mediterranean Diet

Different eating patterns can coexist in the same country. One of them is the Mediterranean diet, which, since November 2010, has been part of the list of Intangible Cultural Heritage of Humanity [35,36]. This diet is characterized by a controlled amount of fat, mainly extra virgin olive oil, a low percentage of carbohydrates, a preference for foods with low glycemic index and rich in dietary fiber, high amounts of natural antioxidants, fish, and moderate consumption of dairy products [37]. Given its components, many research groups have evaluated the relationship between the Mediterranean diet and health. It has been reported that this diet contributes to preventing obesity, metabolic syndrome, and cardiovascular diseases. Thus, the Mediterranean diet has an important role in the quality of life of children and adults [35,38,39].

Cakir et al. analyzed the association between adherence to the Mediterranean diet and the presence of NAFLD in pediatric patients. They considered three groups: a group (*n* = 106) that included obese or overweight children with a recent diagnosis of NAFLD, a second group that included children with obesity without NAFLD (*n* = 21), and a third group (*n* = 54) that included children with normal nutritional status without chronic pathologies [40]. Diet compliance was assessed using the KIDMED index to identify the population with unhealthy eating habits. It was observed that compliance with the diet was good in only 4.7% of patients with NAFLD, while 31.5% in healthy children. In addition, they found that a low score of the KIDMED index is associated with high BMI and obesity in children, being one of the main predictors of NAFLD in children with obesity [40]. Subsequently, Della Corte et al. analyzed the association between adherence to the Mediterranean diet and NAFLD, including laboratory and histological evaluations, in a group of children and adolescents with obesity. They also measured adherence to the diet through the KIDMED questionnaire [37]. The authors found that patients with nonalcoholic steatohepatitis (NASH) had low KIDMED scores, and poor diet adherence was correlated with liver damage (NAFLD activity score >5 and grade 2 fibrosis). In addition, patients with poor diet adherence exhibited higher values of C-reactive protein, fasting insulin, and the HOMA index [39]. On the other hand, Yurtdas et al. [33] and Akbulut et al. [34] verified that the Mediterranean diet (40% carbohydrates, 35–40% fats; <10% saturated fats), 20% protein, and 40–44% carbohydrates, 35–40% fat; <10% saturated fat), 20% protein), for 12 weeks, decreased hepatic steatosis, liver enzymes, and IR in children with NAFLD [33,34]. Therefore, it is inferred that the consumption of this diet should be a protective factor in childhood, mainly in European children. However, it has recently been reported that dietary adherence varies widely within Mediterranean countries in children and adolescents [41], with differences that are also found among several European countries. Furthermore, European diets have changed to the extent of economic growth, incorporating more ultra-processed foods and losing some aspects of the Mediterranean pattern. There are little data available for non-Mediterranean countries, highlighting a study in adolescents from the USA (*n* = 4223) in which it was evidenced that adherence to the diet was low [41,42,43].

**Table 3 nutrients-15-04829-t003:** Studies on dietary interventions and nutritional behavior in children and adolescents with nonalcoholic fatty liver disease.

Intervention	Age Group	Duration	*n*	Results	References
Restricted carbohydrate diet in children and adolescents (Intervention Study).	9–17 years old	8 weeks	25	The intervention had beneficial effects on liver lipids (−6.0 ± 4.7%, by MRI), ↓ ALT (fat restrictive diet: −5.8 U/L; restrictive carbohydrate diet −23.6 U/L), ↓ AST (fat restrictive diet: −2.8 U/L; restrictive carbohydrate diet −20.7 U/L), ↓ body weight (fat restrictive diet: −0.4 kg; restrictive carbohydrate diet −3.0 kg), and IR.	[14]
Diet low in free sugars (glucose, fructose, and sucrose) in male adolescents (Intervention Study).	11–16 years old	8 weeks	20	↓ hepatic steatosis (−6.23%, by MRI-PDFF) and decrease in ALT (−42 U/L).	[17]
Diet low in added sugars in children and adolescents (Intervention Study).	11–16 years old	8 weeks	29	↓ hepatic DNL (34.6% to 24.1%), hepatic steatosis (−8%, by MRI-PDFF), ALT (−30.5 U/L), and ↓ fasting insulin.	[30]
Reduced fructose diet in adolescents (Intervention Study).	11–18 years old	4 weeks	21	↓ hepatic fat (fructose reduction: −0.9%, glucose reduction: 0.2%, by MRS).	[13]
Follow-up of dietary records in a cohort of adolescents (Intervention Study).	14–17 years old	3 years follow up	592	For every 1 g increase in fructose energy-adjusted, the odds of NAFLD in obese adolescents increased by 9% (OR).	[15]
Oral absorption of fructose in obese children with NAFLD (Intervention Study).	8–18 years old	1 day	9	Children with NAFLD absorb and metabolize fructose more efficiently than lean subjects.	[31]
Assess fructose intake in obese children and its relation to NAFLD. (Case-control study).	6–14 years old	Observational study	85	High fructose intake is associated with increased P3NP ( Procollagen type III *N*-terminal peptide) and increased NAFLD grade. P3NP may serve as a marker of NAFLD in obese children with a proposed cutoff value of 8.5 ng/mL	[32]
Diet evaluation in pediatric patients with NAFLD. (Case-control study).	8–18 years old	4 weeks	24	There is a lower consumption of polyunsaturated fatty acids and a higher intake of saturated fatty acids in patients with NAFLD.	[43]
Mediterranean and low-fat diet in children (Intervention Study).	11–18 years old	12 weeks	44	↓ Body weight (−5.1 kg), serum ALT levels (−18 U/L), AST (−10 U/L) and ↓ IR, no significant differences with low-fat diet.	[33]
Mediterranean and low-fat diet in children (Intervention Study).	9–17 years old	12 weeks	60	↓ hepatic steatosis (Mediterranean diet: −0.8%; low fat diet: −0.8%, by ultrasonography), ALT (Mediterranean diet: −25.4 U/L); low fat diet: −24.0 U/L), AST (Mediterranean diet: −10.4 U/L; low fat diet: −15.7 U/L), and ↓ IR improvement.	[34]
Association of the Mediterranean diet and NAFLD in children (Observational study).	8–15 years old	Cross-sectional study	181	Diet compliance is lower in children with NAFLD than in healthy children, a low score on the KIDMED index was associated with a high BMI and obesity in children.	[38]
Association of the Mediterranean diet and NAFLD in children (Observational study).	11–15 years old	Cross-sectional study	113	Diet compliance is lower in children with the pathology than in healthy children; poor adherence to the diet was correlated with liver damage.	[41]
Influence of Chinese dietary patterns on NAFLD among adolescents in Shandong, China (Observational study).	16–23 years old	Cross-sectional study	1639	It was shown that the traditional diet provides a protective effect in preventing NAFLD, while the occidental eating pattern was associated with the pathology.	[42]

Notes: ↓, significant decrease; NAFLD, Non-alcoholic fatty liver disease; AST, aspartate aminotransferase; IR, insulin resistance; MRI-PDFF, magnetic resonance imaging derived proton density fat fraction. MRS, magnetic resonance spectroscopy.

#### 4.3.2. Chinese Diets

In China, Zhen et al. identified two predominant dietary patterns in children and adolescents, namely, the traditional Chinese dietary pattern (with high consumption of rice, vegetables, poultry, pork, and fish) and the modern dietary pattern (with high consumption of wheat, processed meat, and fast food) [44]. They observed that the modern dietary pattern in Chinese adolescents and children is associated with delayed obesity by diet quality and BMI, confirming the importance of culture-specific dietary interventions to reduce obesity rates in children and adolescents [44]. Because of this, Liu et al. studied the influence of dietary patterns and essential foods on NAFLD among adolescents in China, using data taken from the Linyi Nutrition and Health study during 2015 and 2016 that show a prevalence of NAFLD of 13.5% (9.0% men and 4.5% women) [45]. Furthermore, a third eating pattern was incorporated that was classified as high in energy (with high consumption of mushrooms, poultry, fish and shrimp, eggs, fats/oils, nuts, sandwiches, chocolates, and carbonated drinks) [45]. It was also found that children and adolescents with a *“western”* or *“high energy”* pattern had a higher prevalence of obesity, hypertension, and NAFLD, associated with lower family incomes and physical activity levels. Notably, only the Western eating pattern was associated with a higher Odds ratio (OR) for NAFLD (OR = 1.197; 95% CI: 1.013–1.736). The protective effect of the traditional Chinese pattern in preventing NAFLD could be related to the healthy components of this pattern since whole grains contain large amounts of dietary fiber, which is negatively associated with IR, a major risk factor for NAFLD [45]. In addition, specific diet components, such as fish, leafy vegetables, and fruits, are rich in dietary elements (*n*-3 fatty acids) and antioxidant components (vitamins C and E), which have been inversely associated with NAFLD [44,45].

## 5. Nutritional Interventions

### 5.1. N-3 Polyunsaturated Fatty Acids

The intake of *n*-3 long-chain polyunsaturated fatty acids (*n*-3 LCPUFA), such as eicosapentaenoic acid (C20:5n-3, EPA) and docosahexaenoic acid (C22:6n-3, DHA), has been evaluated as a possible treatment for NAFLD [46]. These fatty acids can modulate various mechanisms, including their role in the oxidation of fatty acids, lipogenesis, inflammation, and microbiota modulation [46,47]. In this context, Spahis et al. conducted an RCT in French-Canadian patients between 8 and 18 years old with obesity and a diagnosis of NAFLD. Subjects were given a supplement of *n*-3 LCPUFA (2 g of fish oil per day) in two groups, one for three months and the other for six months, obtaining an increase in plasmatic *n*-3 LCPUFA levels (Table 4) [46]. Moreover, Janczyk et al. evaluated a therapy with DHA and EPA for 24 weeks in patients between 5 and 19 years’ old who presented NAFLD [48]. They found no reduction in ALT levels or hepatic steatosis; however, treated patients exhibited lower levels of AST and gamma-glutamyl transpeptidase compared to placebo [48]. Evaluation of the supplementation with 250 and 500 mg/day of DHA or placebo in Italian children who had liver ultrasonography and biopsy consistent with NAFLD exhibited lower hepatic steatosis and TG, and higher insulin sensitivity with both DHA dosages [49]. In addition, an RCT evaluating supplementation with DHA, choline, vitamin E, and lifestyle changes for 12 months in 60 Italian children diagnosed with NAFLD by biopsy exhibited a significantly decreased hepatic steatosis, ALT, and fasting glucose [50]. To sum up, treatment with DHA and vitamin D in Italian children with obesity, NAFLD, and vitamin D deficiency, including lifestyle changes (weight-decreasing diet and increases in physical exercise) for 12 months, led to a decrease in the NAFLD activity score, serum TG concentration, ALT levels, and IR [51]. 

### 5.2. Amino Acids

There is little evidence in the literature for studying amino acids and NAFLD in children, considering that the liver is the main site of amino acid metabolism. Some metabolomics assays have shown that several amino acid pathways are deregulated in adolescents with NAFLD, tyrosine metabolism being the most affected, along with other routes, including branched-chain amino acids and sulfur-containing amino acids (methionine and cysteine) [52,53].

#### 5.2.1. Cysteine

Reduced glutathione (GSH) is an important intracellular antioxidant in organs, including the liver. GSH depletion has been implicated in developing hepatocellular lesions in hepatic steatosis. GSH is a tripeptide (γ-glutamyl-cysteinyl-glycine) whose absorption from the gastrointestinal tract is low [54]. Thus, ensuring an adequate intake of precursor amino acids, especially cysteine, makes it possible to support intracellular glutathione synthesis. Based on this, Schwimmer et al. sought to determine if 52 weeks’ supplementation with sustained release of cysteamine bitartrate (a cysteine generator) reduces the severity of liver disease in children with NAFLD. Among the findings obtained, there was no histological improvement of the disease, except for a significantly greater recovery in lobular inflammation [54].

#### 5.2.2. L-carnitine

A recent study reported a significant correlation between some acylcarnitines and liver fat content in children and adolescents [41]. Therefore, another amino acid study in children with NAFLD is L-carnitine. This amino acid is an essential nutrient derived from lysine and methionine that plays an important role in lipid metabolism and beta-oxidation of long-chain fatty acids in mitochondria [55]. In that line, Saneian et al. analyzed the effect of L-carnitine supplementation (50 mg/kg/day twice daily for three months) on NAFLD in children and adolescents. This supplementation did not improve biochemical or ultrasonographic markers of NAFLD [55].

### 5.3. Vitamins

Current evidence regarding the role of vitamins in NAFLD in children is scarce. However, it has been reported that some vitamins could potentially improve the parameters affected in NAFLD, mainly because of their action on oxidative stress and fibrogenesis [56,57].

#### 5.3.1. Vitamin E (Alpha-Tocopherol)

Nobili et al. conducted a 12-month RCT involving children diagnosed with NAFLD who underwent nutritional counseling for lifestyle changes plus a placebo or a 600 IU alpha-tocopherol supplement with 500 mg/day of ascorbic acid (vitamin C). It is worth mentioning that vitamin C is administered to enhance the regeneration of vitamin E. The results did not show significant differences between groups concerning ALT, HOMA-IR levels, and weight loss, concluding that lifestyle changes significantly improve liver function independently of vitamin E supplementation [58]. Moreover, an RCT on 173 children and adolescents between the ages of 8 and 17 with a diagnosis of NAFLD confirmed by biopsy were given an 800 IU capsule of vitamin E or 1000 mg of metformin or placebo for 96 weeks. It was concluded that neither vitamin E nor metformin treatments had significant differences in ALT reduction, NAFLD activity score, and other histological evidence (including hepatocellular ballooning, fibrosis, steatosis, and lobular inflammation) [59]. On the other hand, 80 children diagnosed with NAFLD by biopsy were included in an RCT placebo-controlled, and subjects were given a capsule with 7.5 mg of hydroxytyrosol and 10 mg of vitamin E or a placebo capsule for four months. The supplementation resulted in a decrease in IR and TG levels, hepatic steatosis, and an improvement in oxidative stress parameters [60]. Likewise, Mosca et al. conducted an RCT on children with NAFLD diagnosed by biopsy who were treated with two pills containing 3.75 mg of hydroxytyrosol and 5 mg of vitamin E each, or two placebo pills for 16 weeks [61]. At four months, the group with treatment improved the oxidative stress parameters, significantly increasing plasmatic levels of IL-10 and diminishing the levels of interleukin 6, improving IR and steatosis [61]. The study of the effect of a lifestyle intervention alone or combined with a 600 mg/day vitamin E supplement for six months in obese children with hepatic steatosis resulted in a significant improvement in oxidative stress, ALT, lipid profile, and HOMA-IR, suggesting that supplementation with vitamin E achieves these benefits [62]. In the same line is the clinical trial by Zöhrer et al., who assessed a supplement that combined DHA (250 mg/day), vitamin E (39 UI/day), and choline (201 mg/day) in children with NAFLD [50]. The treatment improved hepatic steatosis and diminished ALT and plasmatic glucose levels; however, it is unclear whether this improvement is exclusively associated with one of the three components or the mix [50]. In addition, vitamin E combined with DHA and choline for six months plus six months of follow-up improved radiographic, histological, and hepatometabolic indications of NASH in pediatric patients [63]. According to the studies analyzed, vitamin E can potentially affect the treatment of NAFLD in children through various metabolic effects that reduce steatosis and IR and exert an antioxidant effect. Some of the above trials [50,60,61,63] confirm the contention that the co-administration of two or more cytoprotective agents may control the damage more efficiently, thus avoiding its progression into more unmanageable stages of the disease [64]. This line of reasoning is attributable to agents’ actions through different or similar mechanisms achieving synergistic or additive results and to using lower dosages than the monotherapies and shorter administration periods minimizing side effects [64]. Nevertheless, more studies are required to determine if the outcome is achieved only with the supplementation of this vitamin or if it is an adjuvant with other therapies. 

#### 5.3.2. Vitamin D

Vitamin D has been investigated as a possible treatment for NAFLD due to its anti-inflammatory, antifibrotic, and antiproliferative effects. In addition, NAFLD patients often present vitamin D hypovitaminosis [51,65]. In an RCT in which 109 children with NAFLD confirmed by biopsy participated and were given 2000 IU/day of vitamin D or a placebo for six months, the treated group reduced hepatic steatosis, lobular inflammation, AST, and ALT levels and improved lipid profile and IR [66]. Another study, where children with fatty liver detected by liver ultrasound were randomly assigned to receive a vitamin D supplement (50,000 U/week), found an improvement in the degree of fatty liver by ultrasonography and an increase in blood vitamin D levels, in addition to a significant increase in HDL and a significant decrease in LDL, insulin, HOMA-IR, and ALT [67]. This latest study suggests that vitamin D could be used as an adjunctive treatment for NAFLD in children.

### 5.4. Probiotics

A healthy gut microbiota is essential for the host’s normal intestinal homeostasis development. Recent studies indicate that the disruption of the intestinal microbiota, dysbiosis, could give rise to several pediatric diseases, including NAFLD [60]. The investigations on microbiota changes induced by the administration of probiotics and the association with NAFLD parameters in pediatrics are also scarce [68,69].

#### 5.4.1. Lactobacillus Rhamnosus GG (LGG)

One of the first investigations to evaluate probiotic supplementation in children was a double-blind RCT in which the authors described that obese children with persistent hyper-transaminasemia and bright liver echo pattern on ultrasound treated with LGG for eight weeks resulted in a significant decrease (up to normalization in 80% of cases) in serum ALT values. It is worth considering that one of the limitations was that NAFLD was not confirmed by biopsy [70].

#### 5.4.2. Multi-Strain Probiotics

VSL#3^®^ (Pharmaceuticals Inc., Towson, MA, USA) is a highly concentrated (approximately 5 × 10^11^ cells/g) multi-strain probiotic product containing live freeze-dried lactic acid bacteria and bifidobacteria from 8 different strains (Table 4) [71]. Alisi et al. studied VSL#3 supplementation (1 sachet/day) in biopsy-proven and placebo-controlled obese children with NAFLD [72]. The treatment considered a low-calorie diet and a moderate aerobic exercise program for four months. Under these conditions, a decrease in the severity of the disease detected by ultrasound, a reduction in ALT and IR, an increase in anorexigenic intestinal hormones (glucagon-like peptide 1 and activated GLP-1), and a decrease in BMI in children supplemented with VSL#3^®^ over controls were reported [72]. These effects could be partly explained by restoring the intestinal microbiota that otherwise could result in reduced intestinal permeability, thus contributing to a diminution in the inflammatory state and IR, both preponderant factors in NAFLD [68]. Furthermore, Famori et al. conducted a triple-blind randomized trial in obese children and adolescents with NAFLD, confirmed by ultrasound. Participants were randomly divided into two groups to receive a daily probiotic capsule (Prokid^®^; Gostaresh Milad Pouya Company, Tehran, Iran) or placebo for 12 weeks, along with recommendations for healthy lifestyle habits and physical activity [73]. After the intervention, there was an improvement to normal liver ultrasound in 17 (53.1%) and 5 (16.5%) patients in the intervention and placebo groups, respectively. Moreover, the probiotic group also decreased ALT levels, total cholesterol, LDL, TG, and waist circumference [73]. Another multi-strain probiotic available on the market is Bio-Kult^®^ (Probiotics International Ltd., Somerset, United Kingdom), which contains 14 bacterial strains, 2.2 × 10^9^ colony-forming units per capsule [74]. Thushara et al. recently evaluated the effects of 1 or 2 doses of Bio-Kult^®^ in obese children with ultrasound evidence of NAFLD for six months, including indications for a structured diet and physical activity. The results were inconclusive since the ultrasound revealed an improvement in the NAFLD stage. However, when elastography was performed in a subsample, no significant differences were found [74]. Therefore, the authors conclude that probiotics have no advantage over lifestyle modification in improving the metabolic disorder associated with obesity in children and adolescents [74,75].

**Table 4 nutrients-15-04829-t004:** Dietary therapeutic components in children with nonalcoholic fatty liver disease.

Intervention	Age Group	Duration	*n*	Results	References
Supplementation with *n*-3 LCPUFA (450–1300 mg/day) or placebo in overweight children diagnosed with NAFLD.	Children > 5 years and <19 years old	24 weeks	30	No reduction in ALT levels or hepatic steatosis, ↓ levels of AST (−11 U/L) and GGT (−9 U/L).	[45]
Supplementation with 250 or 500 mg/day of DHA or placebo.	11 to 13 years	6 months	60	↓ steatosis (The odds of more severe versus less severe steatosis after treatment with DHA 250 mg/day (OR = 0.01) and DHA 500 mg/ day (OR = 0.04), by ultrasonography) and TG. It ↑ in insulin sensitivity.	[46]
Delayed-release cysteamine bitartrate supplementation in children with NAFLD, placebo-controlled.	8 to 17 years	52 weeks	88	No histological improvement of the disease, significantly greater improvement for lobular inflammation (−15%), and a decrease in alanine aminotransferase (−45 U/L) and aspartate aminotransferase (−27 U/L).	[51]
Supplementation with L-carnitine 50 mg/kg/day twice a day, in children and adolescents with NAFLD, controlled with a placebo.	5 to 15 years	3 months	31	No improvements in biochemical and ultrasound markers of NAFLD were observed.	[52]
Lifestyle intervention with or without supplementation with 600 mg/day of vitamin E in obese children with liver steatosis.	6 to 10 years	6 months	24	Significant ↓ oxidative stress, ALT (−7.7 U/L), lipid profile and HOMA-IR.	[55]
Supplementation with 2000 IU/day of vitamin D or placebo in children with NAFLD confirmed by biopsy.	9 to 18 years	6 months	55	↓ liver steatosis and lobular inflammation (by liver biopsy), AST (−16 U/L), and ALT levels (−28.4 U/L), ↓ improvement in lipid profile parameters and IR, and ↑ in vitamin D levels (+27.7 ng/mL).	[63]
LGG (1.2 × 10^10^ CFU/day) supplementation in children with obesity, persistent hypertransaminasaemia, and bright liver echo pattern.	10.7 ± 2.1 years	8 weeks	20	↓ ALT (up to normalization in 80% of cases).	[67]
VSL#3 supplementation (1 or 2 sachets) versus placebo in children with obesity and NAFLD confirmed by biopsy.	9 to 12 years	4 months	32	NAFLD improvement (More severe versus less severe steatosis OR: 0.001, by ultrasound), BMI, and ↑ in anorexigenic intestinal hormones (including GLP-1 and GLP-2).	[69]
Probiotic capsule supplementation placebo-controlled for obese children and adolescents with NAFLD confirmed by liver ultrasound.	10 to 18 years	12 weeks	43	Normal liver sonography was reported in 53.1% of patients, ↓ mean levels of ALT (−9.7 U/L), AST (−7.9 U/L), total cholesterol, waist circumference, LDL, and TG.	[70]
Bio-Kult^®^ supplementation in obese children and adolescents with ultrasound evidence of hepatic steatosis and NASH.	5 to 15 years	6 months	84	Probiotics had no advantage over lifestyle modification in improving obesity-associated metabolic disorders in children.	[71]

Notes: ↑, significant increases; ↓, significant decrease; NAFLD, Non-alcoholic fatty liver disease; AST, aspartate aminotransferase; ALT, Alanine aminotransferase; GGT, gamma-glutamyl transferase; IR, insulin resistance; GLP-1, glucagon-like Peptide 1; aGLP-1, glucagon-like Peptide activated; LDL, low density lipoprotein; TG, triglyceride.

## 6. Conclusions

The efficacy of various dietary interventions and nutritional components such as antioxidants, fatty acid supplements, and probiotics in children and adolescents have been investigated. Still, healthy eating and physical activity remain the first-line strategies for NAFLD prevention and treatment in the pediatric population. It should be noted that most dietary and nutritional interventions described in this review article included the prescription of lifestyle changes (healthy diet and exercise) in children and adolescents with NAFLD. Thus, it must be considered that the effects of specific nutritional interventions were not isolated from habit modifications. In addition, regarding the findings reported in the literature, some studies only considered male adolescents; therefore, the efficacy of the same intervention in girls is unknown. Furthermore, some studies have only focused on Hispanic children due to this population’s high prevalence of NAFLD, and it is important to mention that the studies define NAFLD in different ways (liver enzymes, echotomography, computerized axial tomography, resonance, and biopsies) obtaining different results. Likewise, it should be considered that there are preliminary data on different interventions that could offer a healthy effect on children and adolescents with NAFLD. It is worth mentioning that the Mediterranean diet has effects on weight loss, attenuation of lipid profile, hepatic steatosis, and liver enzymes. This diet is low in sugars, moderate in saturated fats, and high in monounsaturated fats, fish, and antioxidants; interventions alone exhibit a therapeutic effect on NAFLD. However, there is no convincing evidence to recommend one nutritional intervention over another since they have several limitations. Although the metabolic alterations associated with NAFLD are significantly reduced, they do not achieve a complete reversal of the disease. Due to the available evidence, the lack of representativeness in gender and ethnicity is a weakness of the present review. Finally, it is imperative to mention that any dietary intervention for treating NAFLD in pediatrics must ensure an adequate diet that avoids reducing the energy required for growth in this age group, and any indication should be made in the context of lifestyle modifications. Studies evaluating long-term follow-up, including larger sample size and covering the most representative pediatric milestones, are needed to assess the efficacy of the lifestyle approach.

## Figures and Tables

**Figure 1 nutrients-15-04829-f001:**
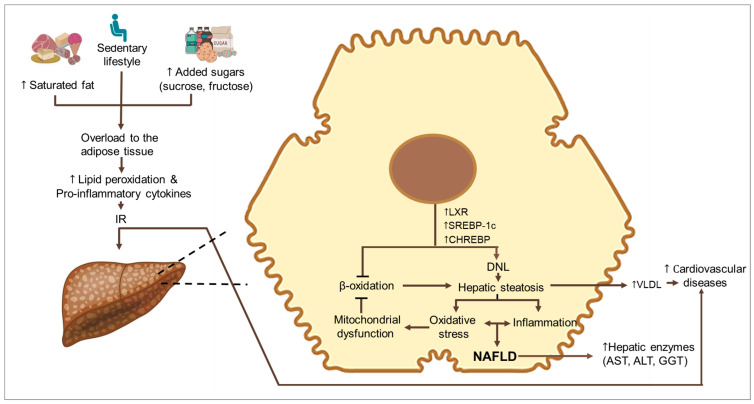
Physio-pathogenesis of non-alcoholic fatty liver disease. IR, insulin resistance; LXR, liver X receptors; SREBP-1C, Sterol-regulatory element binding protein-1C; ChREBP, carbohydrate-responsive element-binding protein, DNL, de novo lipogenesis; NAFLD, Non-alcoholic fatty liver disease, VLDL, very low-density lipoprotein; ALT, Alanine aminotransferase; AST, Aspartate aminotransferase. ↑ Increment.

**Figure 2 nutrients-15-04829-f002:**
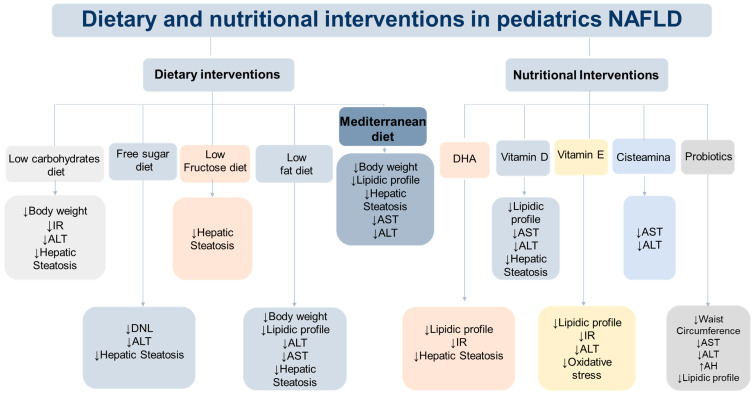
Interventions in nonalcoholic fatty liver disease in children and adolescents. EPA, Eicosapentaenoic Acid; DHA, Docosahexaenoic acid; WC, waist circumference; IR, insulin resistance; TG, Triglycerides; ALT, Alanine aminotransferase; AST, Aspartate aminotransferase; DNL, de novo lipogenesis. ↑ Increment. ↓ reduced.

**Table 1 nutrients-15-04829-t001:** Studies on exercise and dietary changes in pediatric nonalcoholic fatty liver disease.

Intervention	Age Group	Duration	*n*	Results	References
Mediterranean diet and moderate-intensity aerobic physical exercise of at least 60 min, 5 days a week.	In children between 3 and 16 years.	5 days a week.	49	↓ ALT (with BMI reduction: −26.7 U/L).	[20]
Hypocaloric diet (1550 kcal) along with physical exercise for 1 h a day.	In adolescents between 11 and 13 years.	For 10 weeks.	117	↓ 7.1 ± 2.7 kg body weight, an increase in insulin sensitivity, and an improvement in NAFLD parameters (−30% liver echogenicity, −8% had plump liver sign, and −4% changed liver texture by ultrasound).	[21]
Advice on diet restriction and exercise once a week.	In adolescents between 14 and 18 years.	Four months.	52	↓ body fat (−6.0 ± 4.7%) and hepatic lipid content (−32%, by MRI).	[23]

Notes: ↓, significant decrease; NAFLD, Non-alcoholic fatty liver disease; BMI, body mass index; TG, triglycerides; ALT, alanine aminotransferase; AST, aspartate aminotransferase; HDL, high density lipoprotein; IR, insulin resistance.

**Table 2 nutrients-15-04829-t002:** Studies on exercise changes in pediatric nonalcoholic fatty liver disease.

Intervention	Age Group	Duration	*n*	Results	References
Aerobic exercise and aerobic exercise plus resistance exercises for 3 h a week, for one year.	In adolescents between 15 and 19 years.	One year.	68	Greater efficiency in the combination of aerobic exercise plus resistance exercises, achieving a ↓ value of insulin, insulin resistance index (HOMA-IR), and ↓ ALT in aerobic training (–21.84 ± 23.76 U/L) and with aerobic plus resistance training (–5.78 ± 9.73 U/L).	[24]
Low or high-intensity training for 3 h a week for 12 weeks.	In adolescents between 14 and 16 years.	12 weeks.	107	Improvement of the biomarkers of both groups. ↓ ALT, AST (High-Intensity Training: −3.08 and −2.4 U/L, respectively; Low-Intensity Training: −0.9 and −0.9 U/L, respectively) and ↑ 15% HDL of both groups.	[25]
School exercise and high-intensity interval training for 3 days per week for 8 weeks.	In adolescents between 10 and 15 years.	8 weeks.	34	Significant ↓ 1% in body fat, IR, TG, total cholesterol, ALT (−4.4 ± 0.3 U/L), and AST (−3 ± 2 U/L) in both types of training.	[26]
Effect of intense physical activity in a short period of 7 days (3 h per day).	In adolescents between 11 and 13 years.	7 days.	57	Acute weight ↓ (2.53 ± 0.85 kg) and ↓ liver fat content (−1.78 ± 5.53, by abdominal computed tomography) independent of IR.	[27]

Notes: ↑, significant increases; ↓, significant decrease; NAFLD, Non-alcoholic fatty liver disease; BMI, body mass index; TG, triglycerides; ALT, alanine aminotransferase; AST, aspartate aminotransferase; HDL, high density lipoprotein; IR, insulin resistance.

## Data Availability

Not applicable.

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
