# Peer review of "Dietary and Nutritional Interventions in Nonalcoholic Fatty Liver Disease in Pediatrics"

_nutrients, 2023, doi:10.3390/nu15224829_

Round 1

Reviewer 1 Report

Comments and Suggestions for Authors

The paper reviews various dietary interventions for treating NAFLD in children and adolescents. It covers the effectiveness of lifestyle interventions and antioxidants like Vitamin E and D, as well as probiotics.

Key Points:

Vitamin E: Some studies found that Vitamin E supplementation could reduce liver steatosis and insulin resistance (IR). However, results are mixed, and more research is needed to confirm its effectiveness.

Vitamin D: Supplementation has shown promise in reducing liver steatosis and improving lipid profiles. However, more studies are needed.

Probiotics: Limited studies suggest that certain probiotics may improve liver health, but the evidence is not strong enough to recommend them over lifestyle changes.

Lifestyle Changes: Diet and exercise remain the first-line strategies for NAFLD prevention and treatment.

Limitations: Many studies have limitations such as small sample sizes, lack of gender diversity, and short durations.

Conclusion: While some dietary interventions show promise, there is no definitive evidence to recommend one over another. Lifestyle changes remain the most effective treatment.

Strengths of the review:

Comprehensive Review: The paper provides a thorough overview of various dietary interventions for treating NAFLD in children, making it a valuable resource for researchers and clinicians.

Inclusion of Multiple Interventions: The paper does well to include a variety of treatments, from vitamins to probiotics, offering a broad perspective.

Cautious Conclusions: The paper rightly acknowledges the limitations of the existing research and refrains from making strong recommendations, which adds to its credibility.

Weaknesses:

Lack of Clarity in Methods: The methods section of the paper is not sufficiently clear, making it difficult to assess the quality of the review.

Gender and Ethnic Diversity: The paper mentions that some studies only considered male adolescents or focused on Hispanic children. This limitation could be highlighted more prominently as it affects the generalizability of the findings.

Clarity and Structure: The paper is quite dense and could benefit from a clearer structure, perhaps with subheadings for each type of intervention to improve readability.

Discussion on Mechanisms: While the paper discusses the outcomes of various interventions, it could delve deeper into the mechanisms by which these interventions might work.

Omission of Long-Term Ineffectiveness of Low-Carbohydrate Diets: The paper does not discuss the long-term ineffectiveness of low-carbohydrate diets, which is a significant gap given that such diets are often considered for metabolic issues like NAFLD.

Recommendations for Future Research: The conclusion could be strengthened by providing specific directions for future research, given the limitations of the current studies.

Suggestions:

1. Enhance the clarity of the methods section by adhering to the SANRA guidelines—a scale for the quality assessment of narrative review articles. SANRA Guidelines

2. Include a section or subsection that discusses the long-term outcomes of low-carbohydrate diets, particularly their ineffectiveness over extended periods, to provide a more balanced view of dietary interventions for NAFLD.

3. Highlight the limitations regarding gender and ethnic diversity more prominently.

4. Improve the structure for better readability.

5. Include a discussion on the mechanisms of the interventions.

6. Provide specific recommendations for future research in the conclusion section.

Comments on the Quality of English Language

English is fine

Author Response

Santiago, October 24th, 2023. 

EDITORS

NUTRIENTS

Dear Editors,

We really appreciate the peer review of our manuscript entitled “Dietary and Nutritional Interventions in Nonalcoholic Fatty Liver Disease in Pediatrics” nutrients-2670638. In this regard, we have made the suggested modifications. All corrections included are highlighted in yellow in the manuscript.

Reviewer #1

Comments: The paper reviews various dietary interventions for treating NAFLD in children and adolescents. It covers the effectiveness of lifestyle interventions and antioxidants like Vitamin E and D, as well as probiotics.

Key Points:

Vitamin E: Some studies found that Vitamin E supplementation could reduce liver steatosis and insulin resistance (IR). However, results are mixed, and more research is needed to confirm its effectiveness.

Vitamin D: Supplementation has shown promise in reducing liver steatosis and improving lipid profiles. However, more studies are needed.

Probiotics: Limited studies suggest that certain probiotics may improve liver health, but the evidence is not strong enough to recommend them over lifestyle changes.

Lifestyle Changes: Diet and exercise remain the first-line strategies for NAFLD prevention and treatment.

Limitations: Many studies have limitations such as small sample sizes, lack of gender diversity, and short durations.

Conclusion: While some dietary interventions show promise, there is no definitive evidence to recommend one over another. Lifestyle changes remain the most effective treatment.

Strengths of the review:

Comprehensive Review: The paper provides a thorough overview of various dietary interventions for treating NAFLD in children, making it a valuable resource for researchers and clinicians.

Inclusion of Multiple Interventions: The paper does well to include a variety of treatments, from vitamins to probiotics, offering a broad perspective.

Cautious Conclusions: The paper rightly acknowledges the limitations of the existing research and refrains from making strong recommendations, which adds to its credibility.

Weaknesses:

Lack of Clarity in Methods: The methods section of the paper is not sufficiently clear, making it difficult to assess the quality of the review.

Gender and Ethnic Diversity: The paper mentions that some studies only considered male adolescents or focused on Hispanic children. This limitation could be highlighted more prominently as it affects the generalizability of the findings.

Clarity and Structure: The paper is quite dense and could benefit from a clearer structure, perhaps with subheadings for each type of intervention to improve readability.

Discussion on Mechanisms: While the paper discusses the outcomes of various interventions, it could delve deeper into the mechanisms by which these interventions might work.

Omission of Long-Term Ineffectiveness of Low-Carbohydrate Diets: The paper does not discuss the long-term ineffectiveness of low-carbohydrate diets, which is a significant gap given that such diets are often considered for metabolic issues like NAFLD.

Recommendations for Future Research: The conclusion could be strengthened by providing specific directions for future research, given the limitations of the current studies.

Suggestions:

  1. Enhance the clarity of the methods section by adhering to the SANRA guidelines—a scale for the quality assessment of narrative review articles. SANRA Guidelines

Answer:

The methodology section was modified to present more detail. Also, the SANRA scale was applied (see lines 100-103).

  1. Include a section or subsection that discusses the long-term outcomes of low-carbohydrate diets, particularly their ineffectiveness over extended periods, to provide a more balanced view of dietary interventions for NAFLD.

Answer:

We added the following paragraph, starting at line 251: "Nonetheless, the authors indicate that the sample size may not be large enough to determine whether this decrease in liver fat was due to a unique effect of carbohydrate restriction (15). It is important to note that these studies are short-term. A long-term low-carbohydrate diet is difficult to adhere to and could interfere with the recommended intake of fruits and cereals for the pediatric population, which could eventually have negative consequences on development".

  1. Highlight the limitations regarding gender and ethnic diversity more prominently.

Answer:

The following statement was added in line 618: "Due to the available evidence, the lack of representativeness in gender and ethnicity is a weakness of the present review"

  1. Improve the structure for better readability.

Answer:

Thank you for your comment. We grouped the articles analyzed according to the evidence to be classified and provided a structure to the manuscript by the items addressed.

  1. Include a discussion on the mechanisms of the interventions.

Answer:

Some possible mechanisms are discussed throughout the manuscript mode introduction to the subtopic.

Lines 61- 62

Lines 238-242

Lines 286-290

Lines 312-316

Lines 367-374

Lines 431-435

Lines 474-481

Lines 485-489

Lines 532-534

Lines 543-544

Lines 556-560

  1. Provide specific recommendations for future research in the conclusion section.

Answer:

We included the following sentence in line 623: "Studies evaluating long-term follow-up, including a larger sample size and covering the most representative pediatric milestones, are needed to assess the efficacy of the lifestyle approach”.

Reviewer 2 Report

Comments and Suggestions for Authors

Dear Authors,

Congratulations on a fascinating manuscript. The modifications I propose are described below.

Figure 1. The description under the figure should read NAFLD, not NALFLD; the figure should read Cardiovascular diseases and not Diseases cardiovascular.

Page 5 line 16 - please provide the full name of ALT

I propose to add two columns in Table 1 and Table 2, i.e. 1) the age of the patients and 2) the duration of the intervention. This will make the table more readable and easier to find this important information.

In Tables 1 and 2, the "conclusions" column might be better called "results"?

Table 1 publication [20] – please indicate what type of diet it was.

I do not understand what the authors' intention was in the case of Table 1 from section 3.1, because it presents dietary interventions with physical activity (publications from [17] to [20]) and interventions involving physical activity alone (publications from [21] to [24] ) and section 3.2 presents additional publications where physical activity is an intervention. However, they are not included in table 1. Why? Maybe make a separate table for the diet plus activity intervention and a separate table for section 3.2 where the intervention is only physical activity.

Table 2 - I propose to separate studies that looked for a relationship between diet and the risk of developing NAFLD or eating behaviors in patients with NAFLD from studies in which diet was an intervention in patients with NAFLD. You could create a separate table, annotate it before the current section 3.1, and give it a title, "nutritional behavior of pediatric patients with NAFLD". In turn, publication [28] is yet another category.

Table 2 - the description of the publication [29] in the "conclusions" column is unclear: "ˇ

“Decrease in fructose intake is associated with NAFLD.” Was it that a decrease in fructose intake reduces NAFLD?

In individual sections, descriptions of interventions are mixed with assessing certain eating behaviors in patients with NAFLD. Maybe it would be better to sort this out.

Table 3 - please add two additional columns, i.e. age of respondents and time of intervention.

Author Response

Santiago, October 24th, 2023.

EDITORS

NUTRIENTS

Dear Editors,

We really appreciate the peer review of our manuscript entitled “Dietary and Nutritional Interventions in Nonalcoholic Fatty Liver Disease in Pediatrics” nutrients-2670638. In this regard, we have made the suggested modifications. All corrections included are highlighted in yellow in the manuscript.

Reviewer #2

Dear Authors,

Congratulations on a fascinating manuscript. The modifications I propose are described below.

  1. Figure 1. The description under the figure should read NAFLD, not NALFLD; the figure should read cardiovascular diseases and not Diseases cardiovascular.

Answer:

Figure 1 and its description were modified according to the reviewer’s comment.

  1. Page 5 line 16 - please provide the full name of ALT

Answer:

The full name was provided.

  1. I propose to add two columns in Table 1 and Table 2, i.e. 1) the age of the patients and 2) the duration of the intervention. This will make the table more readable and easier to find this important information.

In Tables 1 and 2, the "conclusions" column might be better called "results"?

Table 3 - please add two additional columns, i.e. age of respondents and time of intervention.

Answer:

Tables 1, 2, 3, and 4 were modified according to the reviewer’s comment.

  1. Table 1 publication [20] – please indicate what type of diet it was.

Answer:

This comment was included. “Advice on diet restriction and exercise once a week.”

  1. I do not understand what the authors' intention was in the case of Table 1 from section 3.1, because it presents dietary interventions with physical activity (publications from [17] to [20]) and interventions involving physical activity alone (publications from [21] to [24] ) and section 3.2 presents additional publications where physical activity is an intervention. However, they are not included in table 1. Why? Maybe make a separate table for the diet plus activity intervention and a separate table for section 3.2 where the intervention is only physical activity.

Answer:

This comment was included. Table 2 was created that provides for only physical activity and missing section 3.2 publications.

  1. Table 2 - I propose to separate studies that looked for a relationship between diet and the risk of developing NAFLD or eating behaviors in patients with NAFLD from studies in which diet was an intervention in patients with NAFLD. You could create a separate table, annotate it before the current section 3.1, and give it a title, "nutritional behavior of pediatric patients with NAFLD". In turn, publication [28] is yet another category.

Answer:

We did not generate a new table since studies that evaluated nutritional behaviors are a minority, and most of them are used in the article to provide a context before intervention studies. However, we agree with the reviewer’s comment. Thus, the title of Table 3. was modified to “Studies on dietary interventions and nutritional behavior in children and adolescents with nonalcoholic fatty liver disease”. The type of study (intervention, case control or observational study) is now specified in the intervention column.

  1. Table 2 - the description of the publication [29] in the "conclusions" column is unclear: "ˇ

Answer:

We added a results column in the table (now Table 3).

  1. “Decrease in fructose intake is associated with NAFLD.” Was it that a decrease in fructose intake reduces NAFLD?

Answer:

Thank you for this observation. It was corrected.

  1. In individual sections, descriptions of interventions are mixed with assessing certain eating behaviors in patients with NAFLD. Maybe it would be better to sort this out.

Answer:

The comment is appreciated. Some issues are ethically complex to address through intervention studies, so observational studies allow us to look at certain foods, which is why they were also included in the manuscript.

All authors have read and approved the revised manuscript. We hope our resubmission is suitable for inclusion in Nutrients and look forward to hearing from you.

Sincerely,

Rodrigo Valenzuela, PhD

Department of Nutrition

Faculty of Medicine

University of Chile.

Round 2

Reviewer 1 Report

Comments and Suggestions for Authors

The authors made some improvements to the paper.

However, there remain other issues that I had pointed out that have not been addressed:

Lack of Clarity in Methods: The methods section of the paper is not sufficiently clear, making it difficult to assess the quality of the review.

Gender and Ethnic Diversity: The paper mentions that some studies only considered male adolescents or focused on Hispanic children. This limitation could be highlighted more prominently as it affects the generalizability of the findings.

Clarity and Structure: The paper is quite dense and could benefit from a clearer structure, perhaps with subheadings for each type of intervention to improve readability.

Discussion on Mechanisms: While the paper discusses the outcomes of various interventions, it could delve deeper into the mechanisms by which these interventions might work.

It is very surprising to see low-carbohydrate diets on a par with the Mediterranean diet, particularly in the figure.

Author Response

Santiago, November 06, 2023.

EDITORS

NUTRIENTS

Dear Editors,

We appreciate the peer review of our manuscript entitled “Dietary and Nutritional Interventions in Nonalcoholic Fatty Liver Disease in Pediatrics”. The manuscript presents an updated review of dietary and nutritional interventions in pediatric patients with NAFLD, highlighting the main metabolic outcomes. In this regard, we have made the suggested modifications.

Please find specific responses to comments below.

Reviewer

However, there remain other issues that I had pointed out that have not been addressed:

Answer:

We thank the reviewer for all his comments. Comments that helped us improve the scientific quality of the manuscript. However, we consider it important to mention the following aspects about our review.

  1. The manuscript presents an updated review of dietary and nutritional interventions in pediatric patients with NAFLD.
  2. Our group has published various scientific manuscripts on NAFLD in humans and also in models of hepatic steatosis studying the mechanisms involved.
  3. The manuscript is an update article and we have used the necessary, sufficient and appropriate scientific methodology for this type of scientific manuscripts.

Reviewer

Commentary 1: Lack of Clarity in Methods: The methods section of the paper is not sufficiently clear, making it difficult to assess the quality of the review.

Answer: The methods section was improved (lines 94 – 105). It is worth mentioning that this manuscript is a narrative review and provides an update of existing evidence in this specific population group. Systematic reviews addressing general lifestyle changes (e.g., diet and/or exercise) in pediatrics with NAFLD have been published elsewhere (Abud Alanazi Y. 2023 and Utz-Melere M, et al. 2018). Those reviews do not address specific nutritional interventions.

Commentary 2: Gender and Ethnic Diversity: The paper mentions that some studies only considered male adolescents or focused on Hispanic children. This limitation could be highlighted more prominently as it affects the generalizability of the findings.

Answer: Lines 611-610: emphasis is given to what was stated

Commentary 3: Clarity and Structure: The paper is quite dense and could benefit from a clearer structure, perhaps with subheadings for each type of intervention to improve readability.

Answer: Lines 106 -112: we have added a new paragraph to emphasize exercise and dietary changes, as opposed to physical activity-only interventions. The review is structured according to specific dietary or nutritional intervention subheadings.

Commentary 4: Discussion on Mechanisms: While the paper discusses the outcomes of various interventions, it could delve deeper into the mechanisms by which these interventions might work.

Answer: Thanks for your comments. This review aims to present an updated compilation of the scientific evidence for treating this pathology, highlighting lifestyle and specific nutritional strategies. Mechanisms involved are discussed, but this manuscript does not intend to review molecular mechanisms in depth. Our research group has published various scientific manuscripts on NAFLD in humans and models of hepatic steatosis, studying the mechanisms involved.

We have previously described mechanisms elsewhere:

Echeverría et al., Long-chain polyunsaturated fatty acids regulation of PPARs, signaling: Relationship to tissue development and aging. Prostaglandins Leukot Essent Fatty Acids. 2016;114:28-34.

Valenzuela R, Videla LA. Crosstalk mechanisms in hepatoprotection: Thyroid hormone-docosahexaenoic acid (DHA) and DHA-extra virgin olive oil combined protocols. Pharmacol Res. 2018;132:168-175.

Videla LA, Valenzuela R. Perspectives in liver redox imbalance: Toxicological and pharmacological aspects underlying iron overloading, nonalcoholic fatty liver disease, and thyroid hormone action. Biofactors. 2022;48(2):400-415.

Valenzuela R, Videla LA. Impact of the Co-Administration of N-3 Fatty Acids and Olive Oil Components in Preclinical Nonalcoholic Fatty Liver Disease Models: A Mechanistic View. Nutrients. 2020;12(2):499.

Ortiz M, et al. Suppression of high-fat diet-induced obesity-associated liver mitochondrial dysfunction by docosahexaenoic acid and hydroxytyrosol co-administration. Dig Liver Dis. 2020;52(8):895-904.

Echeverría F, et al. Reduction of high-fat diet-induced liver proinflammatory state by eicosapentaenoic acid plus hydroxytyrosol supplementation: involvement of resolvins RvE1/2 and RvD1/2. J Nutr Biochem. 2019;63:35-43.

Rincón-Cervera MA, et al. Supplementation with antioxidant-rich extra virgin olive oil prevents hepatic oxidative stress and reduction of desaturation capacity in mice fed a high-fat diet: Effects on fatty acid composition in liver and extrahepatic tissues. Nutrition. 2016;32(11-12):1254-67.

Hernández-Rodas MC, et al. Supplementation with Docosahexaenoic Acid and Extra Virgin Olive Oil Prevents Liver Steatosis Induced by a High-Fat Diet in Mice through PPAR-α and Nrf2 Upregulation with Concomitant SREBP-1c and NF-kB Downregulation. Mol Nutr Food Res. 2017;61(12).

Commentary 5: It is very surprising to see low-carbohydrate diets on a par with the Mediterranean diet, particularly in the figure.

Answer: Figure 2 was modified to provide emphasis to the Mediterranean Diet. The following paragraph was added in conclusion: “It is worth mentioning that the Mediterranean diet has effects on weight loss, attenuation of lipid profile, hepatic steatosis, and liver enzymes. This diet is low in sugars, moderate in saturated fats, and high in monounsaturated fats, fish, and antioxidants; interventions alone exhibit a therapeutic effect on NAFLD”.

All the changes are highlighted in yellow in the manuscript.  

We hope your submission is suitable for publication in Nutrients and look forward to hearing from you.

Sincerely,

Rodrigo Valenzuela, PhD

Department of Nutrition

Faculty of Medicine

University of Chile.
